# Maize Production under Drought Stress: Nutrient Supply, Yield Prediction

**DOI:** 10.3390/plants12183301

**Published:** 2023-09-18

**Authors:** Adrienn Széles, Éva Horváth, Károly Simon, Péter Zagyi, László Huzsvai

**Affiliations:** 1Institute of Land Use, Engineering and Precision Farming Technology, Faculty of Agricultural and Food Sciences and Environmental Management, University of Debrecen, Böszörményi Str. 138, H-4032 Debrecen, Hungary; horvath.eva@agr.unideb.hu (É.H.); simon.karoly@simon-kft.hu (K.S.); zagyi.peter@agr.unideb.hu (P.Z.); 2Institute of Statistics and Methodology, Faculty of Economics and Business, University of Debrecen, Böszörményi Str. 138, H-4032 Debrecen, Hungary; huzsvai@econ.unideb.hu

**Keywords:** maize, N fertilisation, LAI, SPAD, yield prediction

## Abstract

Maize yield forecasting is important for the organisation of harvesting and storage, for the estimation of the commodity base and for the provision of the country’s feed and food demand (export–import). To this end, a field experiment was conducted in dry (2021) and extreme dry (2022) years to track the development of the crop to determine the evolution of the relative chlorophyll content (SPAD) and leaf area index (LAI) for better yield estimation. The obtained results showed that SPAD and LAI decreased significantly under drought stress, and leaf senescence had already started in the early vegetative stage. The amount of top dressing applied at V6 and V12 phenophases did not increase yield due to the low amount of rainfall. The 120 kg N ha^−1^ base fertiliser proved to be optimal. The suitability of SPAD and LAI for maize yield estimation was modelled by regression analysis. Results showed that the combined SPAD-LAI was suitable for yield prediction, and the correlation was strongest at the VT stage (R^2^ = 0.762).

## 1. Introduction

Maize is the world’s leading staple cereal crop with an average annual production of 1070 million tonnes (2011–2021) [1,2]. The European Union contributed 72,987 thousand tonnes to world maize production in 2021. Hungary is one of the most important maize producers in the EU, with a production of 6424 thousand tonnes, ranking 4th. However, there are significant extremes in the average yield. Over the last 30 years, the national average yield of Hungary has been 6.05 Mg ha^−1^ with a lower value of 3.7 Mg ha^−1^ and an upper value of 8.63 Mg ha^−1^ [3].

Weather extremes due to climate change pose a challenge. Hungary has an annual mean temperature of 10–11.5 °C, and an average annual precipitation of 500–800 mm (1991–2020), with a significantly different spatial distribution [4]. July is the warmest month of the year with an average of 21.2 °C. Based on climate model results, Hungary will continue to experience a rise in average temperature both on an annual and seasonal basis. By 2050, the annual average temperature is expected to increase by 1–2 °C and by 2071–2100, it is expected to increase 3–4 °C. The average summer temperature increase is 3.5–4.5 °C, reaching up to 6 °C in August. Precipitation is projected to decrease nationally, especially in July and August, but may increase in the northeast and northwest [5].

Constantly decreasing rainfall in summer and more frequent temperature rises lead to more intense summer droughts. Drought is one of the most severe environmental stresses [6,7], causing significant damage to plants. It inhibits growth and development [8,9,10] especially during the silking and grain-filling stages [11]. It affects chlorophyll synthesis, accelerates chlorophyll decomposition and inhibits photosynthetic activity [12,13,14]. It causes an 8–10% reduction in chlorophyll content during the vegetative stage and 18% reduction during the reproductive stage [15,16]. It also alters genetic properties [17] and causes yield reduction [18,19,20,21,22,23]. Maize yield loss can vary from 30 to 90% depending on the intensity and duration of drought stress. Drought stress during the silking and grain filling stages may cause yield losses of 50 and 21%, respectively [24].

Another major limiting factor for maize yield and yield loss is fertiliser use, especially N fertiliser. It is a key element for proper growth and development, is involved in chlorophyll formation and photosynthesis, and is closely related to grain yield [25,26,27,28]. The correct timing of N application improves yield parameters, which increases grain yield [29,30]. However, improper optimisation of the rate and proportion of nitrogen application causes a number of problems in cultivation [31], slows growth rate, reduces grain yield and increases nitrogen loss, thereby affecting environmental quality [32,33,34,35].

Numerous systems have been developed for the N supply of crops, including methods based on chlorophyll measurement, such as SPAD. The principle of the method is that the N status of leaves is reflected in leaf greenness [36]. There is a significant difference in SPAD values for maize of different ages and leaves, with significantly higher values before silking and decreasing values 2 weeks after silking [37]. The correlations between the SPAD value and N content per leaf area and yield are strongly influenced by the crop year, phenological stage, leaf thickness, leaf position and measurement point on the leaf [38,39,40].

Leaf area index (LAI), an indicator of plant growth and physiological development, is highly modified by different stress effects (environmental and technological conditions) [41], especially by the extent of water and fertilizer stress [42]. Water stress reduces LAI, as it limits canopy development by inhibiting leaf number and leaf growth [43]. N deficiency also significantly reduces LAI, as leaves are narrower, resulting in a loss of radiation absorption and utilisation (photosynthesis) [44].

For the dynamic monitoring of yields, the timely observation of plant ecophysiological status (e.g., leaf area, chlorophyll and nitrogen content) and assessment of N requirements have become key to ensure successful maize yields at different growth stages, to improve yield [45,46,47], and to forecast yields [48,49]. 

In the small plot field experiment of the University of Debrecen, the average maize yield in previous years was 14–15 Mg ha^−1^, which is much higher than the yield in these two years. Therefore, maize growers need as much information as possible on how weather and different rates and timing of N fertilisation affect maize yield and the vegetative and reproductive phenophases to make informed decisions.

To achieve this goal, the authors aim to address the following research objectives: (1) What is the effect of drought stress on SPAD and LAI values? (2) In which phenophase does chlorophyll degradation start under drought stress? (3) How does drought affect the uptake of N applied as a top-dressing nutrient? (4) What are the economically optimal fertilizer rates in dry and extreme dry periods? (5) Which variable(s) can be used to most accurately estimate yield? (6) Which phenophase characteristics show the strongest correlation with yield?

## 2. Materials and Methods

### 2.1. Experiment Site

Our experiments were conducted in Hungary, at the Experimental Station of the University of Debrecen (47°33′ N, 21°26′ E, altitude 111 m), in a split-strip-plot field experiment (180 plots) set up in 2011 with two replications of small plots. In the present study, the measurement results of two hybrids (Merida and Armagnac) were evaluated under natural precipitation conditions in 2021 and 2022 (Figure 1).

### 2.2. Soil Data

The soil of the experiment is a deep humic layer of medium-compaction, lowland calcareous chernozem soil (Mollisol-Calciustoll or Vermustoll, clay loam (K_A_ = 42); USDA) with a good condition, an average pH_KCl_ of 6.6, a humus content of 2.6% and a humus layer thickness of 80–100 cm. The soil is calcareous in the upper 80 cm but moderately calcareous (12%) from 100 cm. The original AL-soluble PO_25_ content of the soil is 130 mg/kg, and the AL-soluble K_2_O content is 240 mg/kg.

### 2.3. Experiment Design and Treatments

The fertiliser rates were divided into basal and top-dressing treatments.

Fertiliser treatments:
NameTreatmentA_0_Unfertilised control;A_60_60 kg N ha^−1^ before sowing;A_120_120 kg N ha^−1^ before sowing;V6_90_60 kg N ha^−1^ before sowing + 30 kg N ha^−1^ during the V6 phenophase;V6_150_120 kg N ha^−1^ before sowing + 30 kg N ha^−1^ during the V6 phenophase;V12_120_60 kg N ha^−1^ before sowing + 30 kg N ha^−1^ during the V6 phenophase + 30 kg N ha^−1^ during the V12 phenophase;V12_180_120 kg N ha^−1^ before sowing + 30 kg N ha^−1^ during the V6 phenophase + 30 kg N ha^−1^ during the V12 phenophase.

The maize was sown on 8 April 2021 and 14 April 2022 with a Wintersteiger grain plot seeder. Harvesting of the plots was carried out on 24 September 2021 and 5 October 2022 using a Sampo 2010 plot harvester equipped with an automatic weighing system and an Oros 2011 two-row maize adapter. The plots have 4 rows, row spacing: 76 cm. The area is 27.6 m^2^ (width: 3 m, length: 9.2 m). There are 1.5 m paths between the fertilized plots. Plant density was 73 thousand plants/ha. Harvested grain yield (Mg ha^−1^) was corrected for 14% moisture content. Number of plots involved in the study annually: 36.

### 2.4. Climatic Data at the Experiment Site

An automatic weather station located and operated at the University of Debrecen Experimental Station provided the utilised daily meteorological data. The climate data from the National Meteorological Service of the Debrecen Airport station for the period 1981–2010 were used to show deviations from the multi-year average [4].

Growing degree days (GDD) during the growing season were calculated according to the following formula:(1)$$\mathrm{GDD}=\sum \left(\frac{\left(T_{\max}+T_{\min}\right)}{2}-T_{b}\right)$$
where T_max_ (°C) is the daily maximum temperature, T_min_ (°C) is the daily minimum temperature, and T_b_ (°C) is the base temperature. For a daily mean temperature lower than the base temperature, i.e., if $\frac{\left(T_{\max}+T_{\min}\right)}{2}<T_{b}$, then $\frac{\left(T_{\max}+T_{\min}\right)}{2}=T_{b}$ is used, and the given daily value of the thermal unit of heat is 0 [50]. T_b_ is the temperature below which the rate of development is considered to be 0. In accordance with the technical literature, the heat sum was calculated using T = 10_b_ °C [51,52].

Under natural precipitation conditions, the accumulated Growing Degree Days (GDDs) from post-emergence (VE) to silking (R1) were nearly similar in both years, with 711 °C in 2021 and 707 °C in 2022. The GDD accumulated between R1 and physiological maturity (R6) in 2022 was 771 °C GDD, which was 68 °C higher than in the same period in 2021. In terms of precipitation, 44 mm less precipitation fell between the VE and R1 phenophase in 2022 (51 mm), and 14 mm more precipitation fell between the R1 and R6 growth stage (67 mm) than in 2021.

Overall, the GDD accumulated between VE and R6 was higher in both years (2021: 1414 °C, 2022: 1478 °C) than the multi-year average for the area: 1200–1280 °C [53], and the precipitation was much lower (2021: 148 mm, 2022: 118 mm) compared to the average (346 mm) (Figure 2). The drought in 2022 was exacerbated by the fact that the previous year 2021 was also very poor in rainfall. The year 2022 was the worst drought year in Hungary in the last 40 years.

### 2.5. Measuring Instruments and Test Methods

A Minolta SPAD-502 portable device (Minolta Camera Ltd., Osaka, Japan) was used to assess leaf chlorophyll concentration (SPAD). Several studies have shown that SPAD measurements provide non-destructive, rapid and accurate information on the N and chlorophyll concentration of maize leaves [25,54,55,56]. The instrument is based on the absorption of radiation from leaves in the red (650 nm) and near-infrared (940 nm) range. The calculation is based on the ratio of the intensity of the infrared and red light transmitted through the leaf. This ratio is higher the more red light is absorbed by the leaves of the plant, which is closely related to the chlorophyll concentration. The SPAD value ranges from 0 to over 100 [57]. Measurements were taken per N treatment on the 6th, 7th and 8th plants of the second row from the left of each plot. It was considered that chlorophylls are not homogeneously distributed on the leaf blade [58,59]; thus, measurements were taken at the most suitable location: at the midpoint of the leaf tip and the leaf stem, an equal distance from the leaf margin and the midrib. Between V8 and R1 phenophases, fully developed leaves appeared; between the R1 and R6 phenophases, these were on the leaf next to the ear [60].

In order to quickly and reliably estimate leaf area index (LAI), the SunScan canopy analysis system was applied, which provides information on maize LAI using photosynthetically active radiation (PAR) measured in the plant canopy. The SunScan probe estimates LAI indirectly from measurements of radiation above and below the canopy, which is based on a theoretical relationship between the leaf area and canopy transmittance [61]. The system consists of a probe, a sunlight sensor and a data logger. The 1000 × 13 mm probe contains 64 PAR optical sensors spaced 15.6 mm apart. The data collected by the sensors are stored on the data carrier. Spectral resolution: 400–700 nm. Measurements were evenly spaced between rows 2 and 3. The LAI value for a given plot was calculated from the average of 5 measurements. Measurements were performed in the phenophases V8, V10, V12, Vn, Vt, R1, R3, R6.

### 2.6. Statistical Analysis

Statistical analysis and plots were performed using R 4.3.1. RStudio version 421 was used as the graphical interface [62].

The split-plot design experiment was evaluated using multi-factor analysis of variance. The dependent variables were SPAD, LAI and grain yield. Grouping variables: years (weather), fertilisation and phenophase. The significance level was set at 5%. For significant effects, calculated *p*-values were also given. Multiple comparisons of means were performed using Duncan’s test. The purpose of these tests was to characterise the evolution of the independent variables in the subsequent regression model and the effects of their shaping factors.

Bivariate and multiple linear regression analysis were used for yield estimation. The dependent variable was yield, and the independent variables were SPAD and LAI. The goodness of fit was characterised by the multiple R^2^ and RMSE values. SPAD and LAI values implicitly included the effects of weather, fertilisation and phenophases.

## 3. Results

### 3.1. Chlorophyll Concentration (SPAD)

A pooled analysis of variance of treatments (fertilisation, phenophase and years) showed that all the main factors had an effect (*p* < 0.001) on SPAD. Crop year had the greatest effect, which was followed by phenophase, and fertilisation had the least influence. All the interactions were verified (*p* < 0.001). ANOVA by year at the 0.1% level showed the effect of the main factors (phenophase, fertilisation) on SPAD in both years. Fertilisation had the largest effect on SPAD in 2021 and phenophase in 2022. The phenophase × fertilisation interaction was significant at the 5% level in 2021 and 0.1% level in 2022.

In the year 2021, the highest SPAD values (*p* < 0.05) were detected in different phenophases in the applied fertilizer treatments (Figure 3). The highest SPAD values in the case of the A_0_ treatment were measured in the VT phenopahse (42.4), in the A_60_ treatment, they were measured in phenophase V8 (42.6) in the A_120_ (50.0) and V12_120_ (51.6) treatments, they were measured in phenophase V10, in V6_90_ (51.5) and V6_150_ (55.0) top-dressing treatments, they were measured in phenophase R1, and in V12_180_ treatment, they were measured in phanophase R3 (56.7). There was no significant change compared to the verified maximum SPAD values for the R1 phase, while a decrease in SPAD value was verified at the 5% level for the R6 phenophase. Top-dressing treatments showed the largest decrease of around 30%. In the studied growing period (V8-R6), the A_0_ treatment had the lowest SPAD value (39.2), while in comparison, the highest SPAD value was ensured by the treatment with 120 kg N ha^−1^ applied as basal fertilizer with an additional 30 kg N ha^−1^ fertilizer dose in the V6 phenophase (V6_150_) (48.5; *p* < 0.05), with an increase of 23.7%.

In the year 2022, the highest SPAD value (*p* < 0.05) was detected in the V8 phenophase (*p* < 0.05) in all fertilizer treatments except for the 60 kg N ha^−1^ treatment (A_60_) (Figure 3). For the R1 phenophase, the A_0_ treatment showed the largest reduction of 23.7% (*p* < 0.05), while for the R6 phenophase, all treatments showed a reduction of 76–79% (*p* < 0.05). For the studied growing period (V8-R6), a 20% increase in SPAD was achieved with the lowest 60 kg N ha^−1^ basal treatment (A_60_; *p* < 0.05) compared to the very low SPAD value of the A_0_ treatment (32.0). The higher 120 kg N ha^−1^ (N_120_) treatment or the application of additional top dressing did not significantly increase SPAD.

Averaged over two years, the highest SPAD values were detected in the V8 phenophase in treatments A_0_ (41.4), A_120_ (46.3) and V12_180_ (46.5) (*p* < 0.05), while the highest SPAD values were detected in treatments A_60_ (46.9), V6_90_ (48.2), V6_150_ (50.5) and V12_120_ (50.7) top dressing in the V10 phenophase (*p* < 0.05). For R1 phenophase, compared to the highest SPAD value verified at the 5% level, the lowest SPAD value reduction was detected in the A_120_ treatment (1.9%; not significant), and for R6 phenophase, it was detected in the V12_180_ (47.1%; (*p* < 0.05) treatment. For both phenophases (R1, R6), the largest reduction was detected in the V12_120_ top-dressing treatment (8.3%, *p* < 0.05; 52.9%, *p* < 0.05). For the growing period (V8-R6), A_0_ treatment had the lowest SPAD value (35.6), with A_60_ providing the highest SPAD value at the 5% level in comparison, with an increase of 13.5%.

The crop year had a significant effect (*p* < 0.001) on the relative chlorophyll content (SPAD). The average SPAD value for the two dry years was 40.1. In 2021, an above average SPAD value of 43.8 was detected, while in the exceptionally dry year of 2022, a critically low SPAD value of 36.4 was measured, with the difference verified at the 0.1% level. In 2022, the average SPAD value of fertiliser treatments was lower (*p* < 0.001) than in 2021 for all development stages except the V8 phenophase. The results obtained support the findings of [63] and [64] that drought limits photosynthesis, water and nutrient uptake, thereby reducing development. The largest difference between the examined years was observed in the R3 and R6 phenophases, representing a reduction of SPAD values of 14.4 (*p* < 0.001) and 24.6 (*p* < 0.001), respectively. Averaged over the examined phenophases (V8-R6), all fertiliser treatments had significantly lower SPAD values in 2022. The smallest difference was found in the V6_90_ (7.0; *p* < 0.05) treatment and the largest was found in the V6_150_ (9.8; *p* < 0.001) treatment (Figure 3).

### 3.2. Leaf Area Index (LAI)

In this study, the authors investigated the interaction of phenophase, fertiliser, and crop year and the interaction of these factors on the LAI of maize. The pooled analysis of variance (ANOVA) and the yearly performed ANOVA showed that all main factors had an effect (*p* < 0.001) on maize LAI. The interaction between phenophase and fertiliser application was confirmed in both years (2021 *p* < 0.05; 2022 *p* < 0.001).

LAI dynamics of maize plants varied each year and in each fertiliser treatment (Figure 3). Evaluating the fertilizer treatments, it was confirmed that in 2021, except for the V12_120_ treatment (V12), the maximum LAI was detected in the R1 phenophase (*p* < 0.05), while in 2022, in all treatments, it was detected in the Vn phenophase (*p* < 0.05). Then, as the leaves aged, it gradually decreased by the time of the phenophase of physiological maturity (R6). The decrease in LAI was lower in 2021 (41.1%) than in 2022 (50%), confirming the result of [65] that leaf maturity depends on LAI. In both growing seasons (V8-R6), the 120 kg N ha^−1^ (N_120_) treatment applied as basal fertilizer had the maximum LAI value (2.48 and 1.73). Compared to the non-fertilized treatment (A_0_), the LAI increase was 58.0% (*p* < 0.05) in 2021 and 39.5% (*p* < 0.05) in 2022 (*p* < 0.05).

Evaluating the average of two years, the highest LAI value, with the exception of A_120_ and V12_120_ treatments, was detected in the Vn phase. In the studied growing period (V8-R6), the A_0_ treatment had the lowest LAI value (1.41), while in comparison, the highest LAI value (2.10; *p* < 0.05) was detected in the case of the treatment with 120 kg N ha^−1^ fertilizer (A_120_) applied as a base fertilizer, with an increase of 48.9%.

Of the three main factors, ANOVA mean squares (MS) data showed that the crop year had the greatest influence on the LAI of maize leaves. Averaged over the two years, the main mean value of LAI was 1.74, increasing by 14.9% in 2021 and decreasing by 26.5% in 2022. Regarding the phenophases, LAI was lower in all phenophases in 2022, but this difference was not significant in the V10, Vn and VT development stages. The largest difference (1.57) was detected in the silking (R1) period (*p* < 0.001). The LAI-reducing effect of the extremely dry year was reliably detected in all treatments except A_60_. The largest differences were observed for the effects of the V12_120_ (0.88; *p* < 0.05) and V12_180_ (0.72; *p* < 0.01) top-dressing treatments.

### 3.3. Grain Yield

The pooled results of ANOVA showed that maize grain yield was influenced by two main factors, year (*p* < 0.001) and fertiliser application (*p* < 0.001), with the effect of crop year being more significant. Of the interactions, year x fertilisation was significant (*p* < 0.001); i.e., grain yield was differently affected by year and fertilisation. A significant (*p* < 0.001) effect of fertiliser application on grain yield was also found in the analysis of variance performed each year.

Yield loss depends on the development period of the crop and the length of the dry period [66,67,68]. The dry period around R1 (silking) is the most sensitive period for maize [69,70,71], which is confirmed by the authors’ own results. Between the V6 and V12 phenophases in 2021, only 10 mm of precipitation fell, and from the V12 phenophase to the R1 (silking) period, 18 mm of precipitation was recorded. The drought in 2021 continued in 2022. Between phenophases V6 and V12, 19 mm of precipitation fell, while between V12 and R1, there was no precipitation.

In both years, due to dry conditions, nitrogen applied in the V6 and V12 phenophases resulted in lower grain yields (2021:10.830 Mg ha^−1^; 2022:7.476 Mg ha^−1^) than a single pre-sowing application (120 kg N ha^−1^) of nitrogen (*p* < 0.001), demonstrating the importance of matching water and nitrogen supply levels [72,73,74,75]. Compared to the non-fertilised (A_0_) treatment (2021: 6.056 Mg ha^−1^; 2022: 4.101 Mg ha^−1^), the A_120_ fertiliser treatment resulted in a 90.7% yield increase in 2021 and 117% in 2022 (Figure 4 and Figure 5).

In 2022, due to a more severe drought, the yield reduction was shown at all nutrient levels (*p* < 0.001) compared to 2021. In the non-fertilised (A_0_) and the V6_90_ and V12_180_ top-dressing treatments, the rate of reduction exceeded 30%. The highest yield reduction was 36.3% in the V12_180_ treatment, while the lowest was 12.1% in the 60 kg N ha^−1^ treatment applied before sowing (A_60_).

### 3.4. Correlation Analysis

Several studies confirm the correlation between chlorophyll concentration and nitrogen [76,77,78,79,80]. When examining the correlation between SPAD and yield, the best fit was obtained for the R3 phenophase (R^2^ = 0.716) (Table 1). In the R6 phase, on the other hand, this relationship virtually disappears. The analysis shows that from the V8 phenophase, the correlation becomes stronger until the R3 phase (Figure 6). The obtained results confirm the research of [81] that the longer maize remains green and photosynthesis is uninterrupted the higher the yield.

Analysis of the correlation between LAI and yield showed a significant positive correlation, which is in agreement with the findings of several researchers [82,83,84,85]. The closest correlation was found in the VT phenophase (R^2^ = 0.627) in contrast to the findings of [86] who found the closest correlation in the R1 phenophase in their experiment. After the VT phase, the relationship between LAI and yield decreased (Figure 7).

Since the SPAD value is related to chlorophyll concentration, it is not by itself an appropriate indicator of the amount of organic matter incorporation. It is necessary to know the surface area that is carrying out photosynthesis, i.e., LAI. The largest amount of biomass is obtained when the values of SPAD and LAI are as high as possible. To assess the combined effect of SPAD and LAI, an estimator model was developed. The estimator model was constructed with Occam’s principle in mind. With few parameters, we aimed to obtain a yield prediction with reasonable accuracy at the earliest possible phenophase. After plotting the measured data, the correlation between SPAD and yield and LAI and yield was found to be linear. By including both explanatory variables in the model, a regression plane was obtained, the parameters of which were estimated using a multiple linear regression model. On the basis of the measurements carried out, analysing the joint effect of SPAD and LAI on yield, the closest correlation was obtained in the VT phenophase (R^2^ = 0.762). The average error of the function fit is ±1.4 Mg ha^−1^. The goodness of fit is only slightly smaller in the Vn phase.

The multiple linear regression model estimates the expected yield in the VT phase with appropriate accuracy.

The estimation model in the VT phenophase (Figure 8):Yield (Mg ha^−1^) = −7.87 + 2.88 × LAI + 0.24 × SPAD(2)

## 4. Discussion

Climate change will affect all regions of the world. There will be areas where precipitation will increase or decrease and evapotranspiration rates will increase, resulting in increased drought [87,88,89]. This means that drought and severe water shortages will negatively affect nutrient uptake [75,90,91], which is one of the causes of yield losses.

Relative chlorophyll content (SPAD) and leaf area index (LAI) provide important information on the nutritional status of plants. Monitoring these indicators is crucial for nutrient replenishment, yield and yield prediction in precision agriculture [42,92,93].

The SPAD value was highest in all fertiliser treatments, averaged over the two dry years, at the early vegetative development stage (V8–V10). Due to drought, chlorophyll degradation and leaf senescence started [94,95,96], then gradually decreased and reached its lowest value at phenophase R6. In 2021, due to the performed top-dressing fertilisation treatments, it gradually increased with the progression of the phenophases and reached its maximum value at the early reproductive stage (R1–R3), which was followed by a decrease (35.9–39.2). In the extreme drought year of 2022, SPAD values for all nutrient levels except treatment A_60_ were at their maximum at the V8 development stage; then, they gradually decreased and showed a minimum value (10.0–11.8) for the R6 phenophase. In these two dry years, with two exceptions (treatments V6_150_ and V12_180_ in 2021), the relative chlorophyll content (SPAD) did not reach the maximum range of 52–56 SPAD values recommended by [36].

The annual analysis showed that in both years, treatment A_0_ resulted in the lowest SPAD value. The obtained results confirmed the findings of [90,97] that drought reduces chlorophyll concentration in maize leaves, thereby reducing nitrogen concentration. Of the two dry years with slightly more rainfall and better rainfall distribution (2021), the top dressing (V6_150_) had an increasing effect on SPAD, while in the severe drought year (2022), the lowest dose rate of the basal treatment (A_60_) had an increasing effect on SPAD.

The leaf area index (LAI) was differently affected by water (precipitation) and fertiliser in the studied dry and extreme dry years. The LAI response to water deficit was smaller in 2022 than in 2021. Water deficit reduced LAI to different extents, which was similar to the results of [98]. Due to water deficit, the optimal maximum leaf area index of maize, which is 4.1–5.9 in Hungary, could not be reached in any year due to fertilisation [99,100]. Top-dressing application did not increase LAI; the basic 120 kg N ha^−1^ treatment (A_120_) provided the maximum LAI value. The results obtained confirmed the finding [101] that low LAI, indicative of low solar energy use efficiency, is dependent on nitrogen availability [42,102], and low grain yield can be expected as a result.

Water shortages were significant in both years at critical phenophases and throughout the growing season. At 120 kg N ha^−1^ (A_120_), the basal fertiliser provided the highest grain yield, while drought stress inhibited further N uptake applied as top dressing in a similar manner [90], leading to grain yield reduction. Water shortages were even more severe in 2022 than in 2021, causing yield reductions of more than 30% in the non-fertilised and top dressing treatments, which was consistent with the results of [103]. This result confirms previous research [104] that further increases in nitrogen fertiliser did not alleviate drought stress.

The timely estimation of SPAD and LAI values is of great importance in precision agriculture. The applied Soil and Plant Analysis Development (SPAD) and SunScan canopy analysis system have been found to be suitable for yield prediction, similarly to several researchers [105,106,107]. We agree with the findings of researchers that the measurements are time consuming, labour intensive and not suitable for large area measurements. The future lies in remote sensing by unmanned aerial vehicles (UAV) as opposed to manual sensors. UAV technology has several advantages, such as fast data acquisition, it is less affected by weather, and it has high spatial and temporal resolution. It has the disadvantages of high investment cost (its professional use requires expertise (training or hiring an expert), which is costly) and short flight time [105,106,107].

Due to the simplicity of our model, it is well applicable in practice, as only SPAD and LAI values are required and an accurate estimate of the yield for a given year can be provided already at the VT stage. A limitation of the model may be the need to have measuring instruments with sufficient accuracy to continuously monitor the development of maize and to have sufficient measurements in the VT phase to make a reliable forecast.

## 5. Conclusions

Drought stress significantly reduced SPAD and LAI values, and chlorophyll decomposition and leaf senescence started already in the late vegetative stage; it also inhibited the uptake of N applied as top dressing. In dry years, 120 kg N ha^−1^ applied as a basal fertiliser proved to be the economically optimal fertiliser rate. The most accurate prediction of maize yield was in R3 (R^2^ = 0.716) in the case of SPAD and VT (R^2^ = 0.627) in the case of LAI. The combination of SPAD and LAI within the introduced model provided the most accurate yield prediction (VT, R^2^ = 0.762), which was non-destructive and cost-effective even in dry and extreme dry years. The combined use of the sensors allows farmers to plan their farming in a controlled way and to react in a timely manner to detected problems. Furthermore, it is assumed that by adapting UAV and handheld sensors together, crop growth monitoring and yield forecasting can be further refined; also, further research is required to validate and develop this method.

## Figures and Tables

**Figure 1 plants-12-03301-f001:**
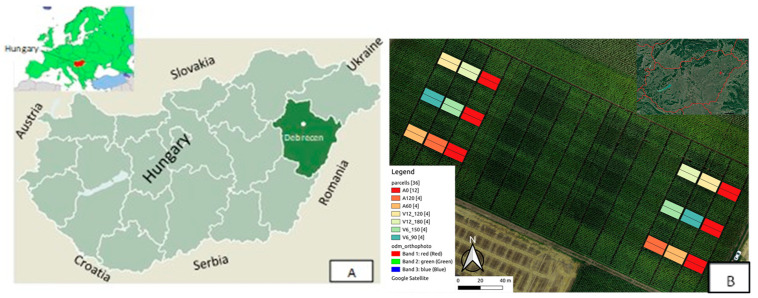
Location of the experimental area. ((**A**): Hungary, Debrecen; (**B**): layout of the basal and top-dressing fertilisation experiment. Measurements were conducted on plots marked with different colours).

**Figure 2 plants-12-03301-f002:**
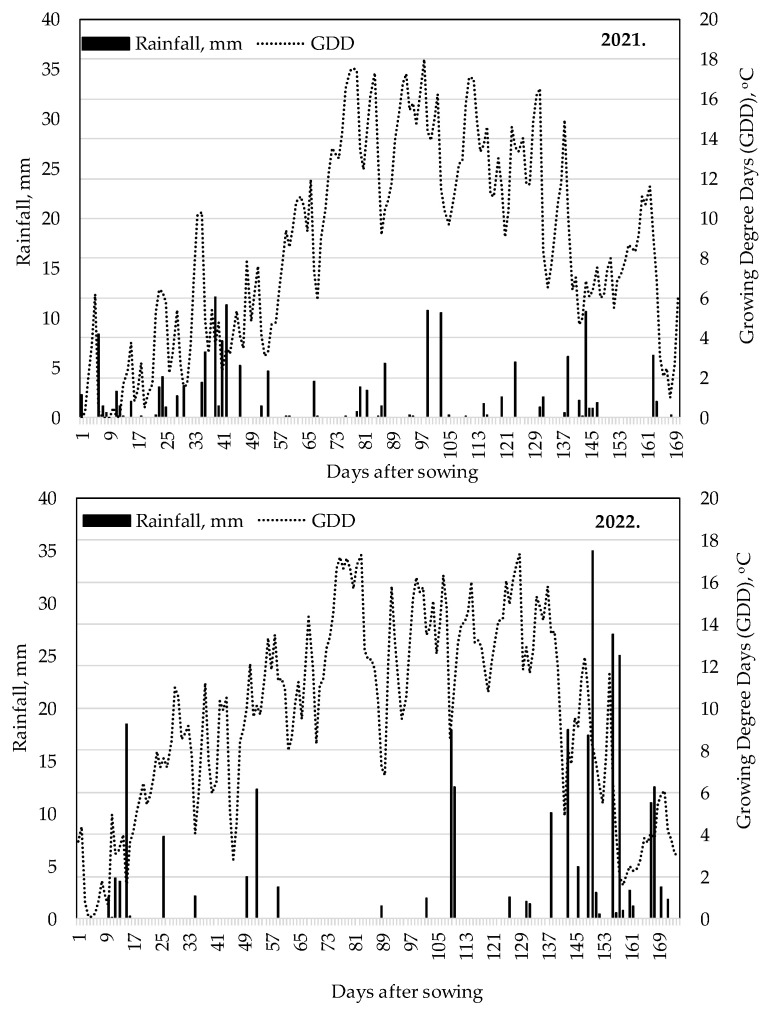
Growing Degree Days and precipitation distribution during the vegetation period, Debrecen, 2021 and 2022.

**Figure 3 plants-12-03301-f003:**
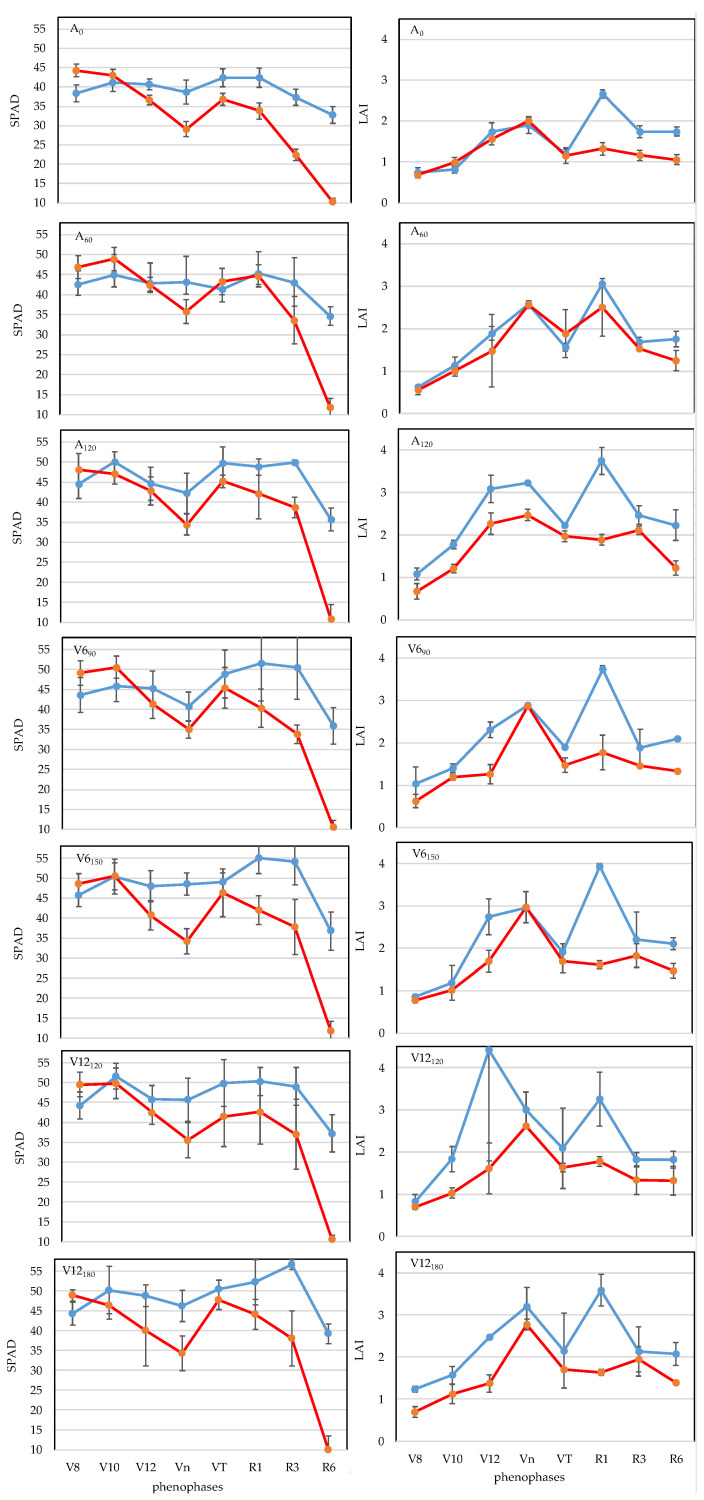
Changes in relative leaf chlorophyll content (SPAD) and leaf area index (LAI), Debrecen, 2021 and 2022. Note: SPAD (**left panel**), LAI (**right panel**), year 2021 is shown in blue, year 2022 is shown in red. Basal and top dressing treatments are shown in rows A_0_ to V12_180_.

**Figure 4 plants-12-03301-f004:**
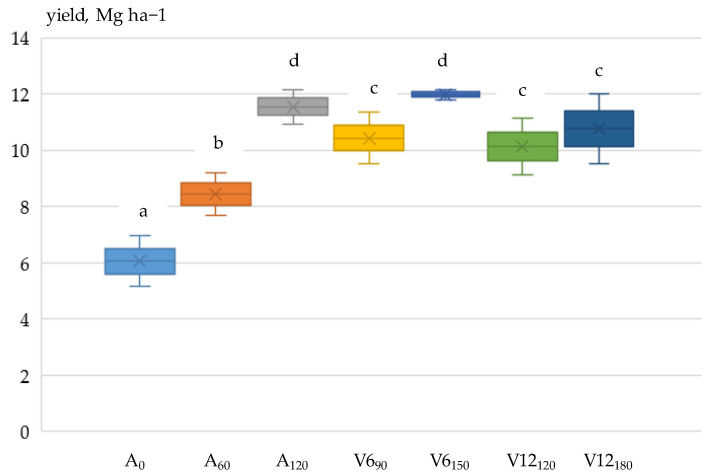
Effect of basal and top-dressing fertilisation on maize yield, Debrecen, 2021. Note: The fertiliser treatments of the different values in lower case are significantly different from each other according to Duncan’s test at the *p* < 0.05 probability level.

**Figure 5 plants-12-03301-f005:**
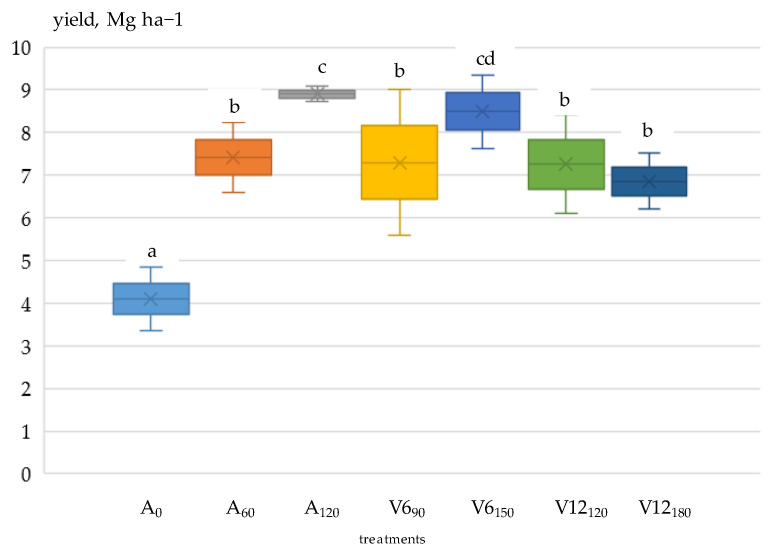
Effect of basal and top-dressing fertilisation on maize yield, Debrecen, 2022. Note: The fertiliser treatments of the different values in lower case are significantly different from each other according to Duncan’s test at the *p* < 0.05 probability level.

**Figure 6 plants-12-03301-f006:**
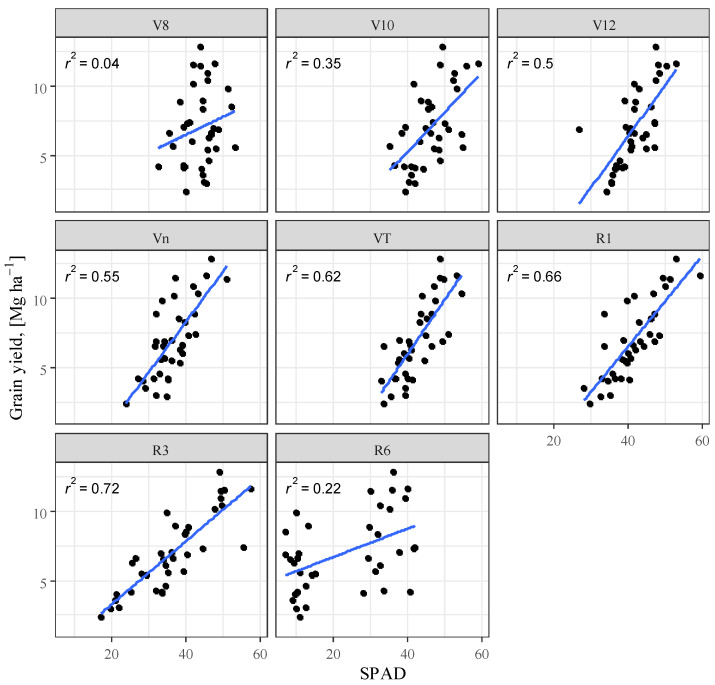
Maize yield as a function of relative chlorophyll content (SPAD), Debrecen, 2021–2022.

**Figure 7 plants-12-03301-f007:**
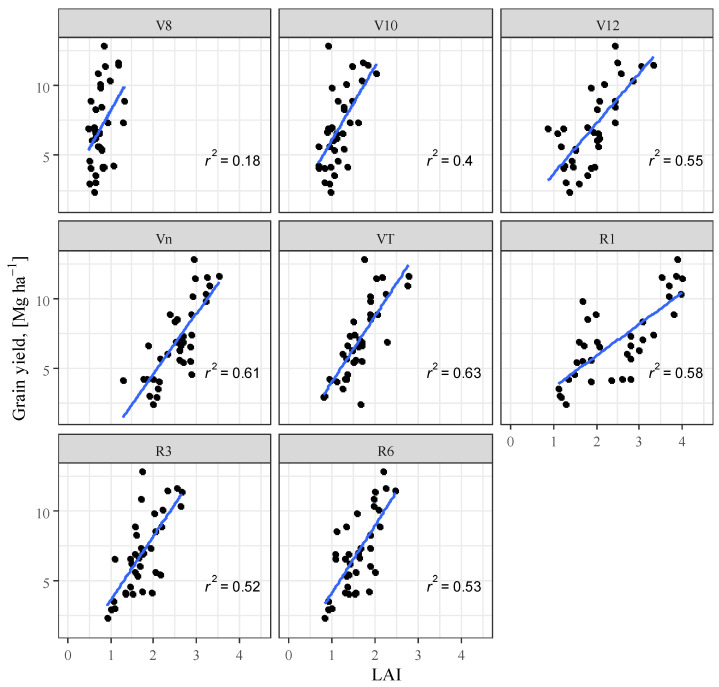
Maize yield as a function of leaf area index (LAI), Debrecen, 2021–2022.

**Figure 8 plants-12-03301-f008:**
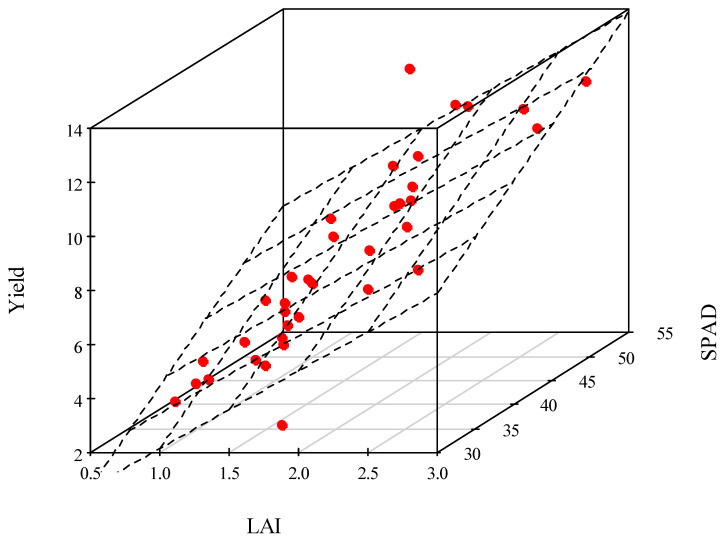
Maize yield as a function of relative chlorophyll content (SPAD) and leaf area index (LAI) in the VT phenophase, Debrecen, 2021–2022.

**Table 1 plants-12-03301-t001:** Multiple R-squared and residuals standard error (RMSE).

Maize Phenophase	R^2^ (SPAD)	R^2^ (LAI)	R^2^ (SPAD + LAI)	RMSE (SPAD + LAI)
V8	n.s.	0.77	0.267	2.469
V10	0.352	0.396	0.475	2.089
V12	0.500	0.226	0.515	2.008
Vn	0.545	0.611	0.734	1.486
VT	0.624	0.627	0.762	1.408
R1	0.655	0.575	0.704	1.569
R3	0.716	0.521	0.733	1.489
R6	0.221	0.526	0.562	1.908

## Data Availability

All data supporting the conclusions of this article are included in this article.

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
