# Peer review of "Maize Production under Drought Stress: Nutrient Supply, Yield Prediction"

_plants, 2023, doi:10.3390/plants12183301_

Round 1

Reviewer 1 Report

The authors made a lot of effort but did not discuss many aspects. I believe they should refer to similar publications such as https://doi.org/10.3390/rs14051281 The article is about a digital elevation model (DEM) is an essential element of input data in the model research of watersheds. The study, gradually and with various methods, carried out a great simplification of a detailed LiDAR-derived DEM. Then, the impact of that treatment on the precision of the selected elements for modeling a watershed was assessed. The simplification comprised a reduction in resolution, with the use of statistical resampling methods, namely giving an average, modal, median, minimum, maximum, or the closest value to the pixels. 

English very difficult to understand/incomprehensible

Reviewer 2 Report

1. The introduction should include a discussion of the current research progress, the existing state of the field, and the limitations of existing studies. This article has introduced the instrument in Section 2.5, and it is suggested to delete the introduction about the sensor in the introduction.

2. Please explain in the text how the corn is sown, the size of the test plot (length x width), and whether different fertilizer treatment plots are treated accordingly to prevent fertilizer from spreading between plots. Please mark the plots used in this experiment in Figure 1. What is the total number of plots used in this study? And the name of the species needs to be explained. Lines 79-81 describe the design of all plots in the test site or the plot design of this study? It is recommended to introduce only the plots. used in this study, and it is suggested to rearrange the expression.

3. The content in the Results section appears to be quite detailed and could be simplified for clarity. Please consider organizing and simplifying the presentation of results.

4. The title of the article is about yield estimation using sensors, but there is limited information on the model establishment in the paper. It is suggested to optimize the title. Additionally, the yield estimation model developed in the study (lines 399-403) and the experimental results should be presented in the Conclusion section of Chapter 3, rather than in the Discussion section.

5. When estimating corn yield using handheld sensors in field production, are there drawbacks such as high workload and low efficiency? In comparison with using unmanned aerial vehicle (UAV) remote sensing platforms for yield estimation, what are the advantages and limitations of handheld sensor-based estimation? The author should emphasize the applicability and limitations of the model in the Discussion section, while also providing suggestions for applying the model to actual agricultural production.

6. Corresponding conclusions should be provided in the Conclusion section for the research objectives outlined in the Introduction.

7. Line 338 in Table 1 has abbreviated residuals standard error as (RMSE), so the unit in the table should use the abbreviation. Also, it is recommended that the units in the table be separated with () or/with SPAD or LAI.

Minor editing of English language required

Reviewer 3 Report

The manuscrip develops how two methods SPAD and LAI can predict maize yield in different climatic conditions and under different fertilization rates.

The introduction is well written and provides a good overview of the state of the art. The material and methods is complet.

The results are well presented, only figure 3 arrives a little too late in the text.

The discussion is wide and interesting.

Conclusions are fine

Minor corrections can be found directly highlighted in the text in the document attached

Round 2

Reviewer 1 Report

There are a number of deficiencies in the authors' manuscript, including the lack of novelty and interesting results. The current form of this article may not be suitable to be published in its entirety. It would be beneficial to the authors if they revised the manuscript carefully in order to improve its novelty, as well as ensure that it meets the standards for research articles.

There are many unfinished points throughout the manuscript. The entire article is about a few pages of content, but nothing innovative is written in it. The manuscript has a big lack of topic knowledge.

No research on it has helped to enrich science. The article is very little research and is not suitable for publication in this state, it would be necessary to thoroughly rebuild all parts of the manuscript.

The discussion was not conducted by the guidelines of the journal. An expansion of literature, conclusions, and discussions is required.

In this form, I advise against publishing the article in this journal.

English needs a lot of improvement.
